# Uncultured Microorganisms and Their Functions in the Fermentation Systems of Traditional Chinese Fermented Foods

**DOI:** 10.3390/foods12142691

**Published:** 2023-07-13

**Authors:** Jiaxuan Wang, Shuyue Hao, Qing Ren

**Affiliations:** China Food Flavor and Nutrition Health Innovation Center, Beijing Technology and Business University, Beijing 100048, China; wangjiaxuan6868@163.com (J.W.); haoshuyue412@163.com (S.H.)

**Keywords:** traditional Chinese fermented foods, uncultured microorganisms, microbial diversity, high-throughput sequencing, pure culture methods

## Abstract

Traditional Chinese fermented foods are diverse and loved by people for their rich nutrition and unique flavors. In the fermentation processes of these foods, the microorganisms in the fermentation systems play a crucial role in determining the flavor and quality. Currently, some microorganisms in the fermentation systems of traditional Chinese fermented foods are in a state of being unculturable or difficult to culture, which hinders the comprehensive analysis and resource development of the microbial communities in the fermentation systems. This article provides an overview of the uncultured microorganisms in the natural environment, in the fermentation systems of traditional Chinese fermented foods, and the research methods for studying such microorganisms. It also discusses the prospects of utilizing the uncultured microorganisms in the fermentation systems of traditional Chinese fermented foods. The aim is to gain a comprehensive understanding of the microbial diversity and uncultured microorganisms in the fermentation systems of traditional Chinese fermented foods in order to better exploit and utilize these microorganisms and promote the development of traditional Chinese fermented foods.

## 1. Introduction

Traditional fermented foods in China have a long history and a wide variety, occupying an important position in the food industry. Based on the sources of the raw materials, traditional Chinese fermented foods can be divided into fermented soybean products, fermented grain products, fermented dairy products, fermented meat products, fermented vegetable products, and fermented tea products. These fermented foods, developed over thousands of years, have formed unique production methods and flavor characteristics, and they are an important part of Chinese culinary culture.

Fermented foods are a category of nutritionally rich and uniquely flavored foods that are produced through microbial fermentation. During the fermentation process, enzymes from microbial sources degrade the macromolecular substances (such as starch, proteins, cellulose, polysaccharides, etc.) in the raw materials into small molecular nutrients (such as monosaccharides, amino acids, etc.). These small molecular substances are utilized by the fermenting microorganisms to produce end products such as ethanol, lactic acid, acetic acid, etc., accompanied by numerous flavor compounds [1,2]. Traditional Chinese fermented foods adopt naturally occurring fermentation, where the microorganisms mainly come from the raw materials, fermentation starters, utensils, and fermentation environment, forming complex microbial ecosystems. Within the same category of fermented foods, the microbial ecosystems differ in different regions of China, resulting in the diverse qualities and flavors of fermented products. For example, Chinese Baijiu (white liquor) has 12 aroma types. This demonstrates that the microbial species and microbial ecological structures in the fermentation systems are the main factors determining the quality and flavor of fermented foods. Studying the microbial diversity in fermented foods and their fermentation systems, isolating pure cultures of microorganisms, and exploring their biological functions are of great significance [3]. However, due to limitations such as the narrow culture conditions and methods, some microorganisms still remain unculturable or difficult to culture, greatly restricting the exploration and research of their potential functions. Therefore, finding effective methods for culturing and reviving unculturable microorganisms, investigating unculturable microorganisms in the fermentation systems of traditional Chinese fermented foods, and uncovering their functionalities will facilitate the development of more beneficial microbial products for industrial production and daily life.

## 2. Uncultured Microorganisms in the Natural Environment

Microorganisms are the most widely distributed and abundant group of organisms on Earth, and the natural world harbors abundant microbial resources that are closely related to human life. On the one hand, microbial resources are the foundation of life science development. Modern molecular biology is based on research into microorganisms such as Escherichia coli. Without in-depth research on microorganisms, the development of modern life science would not have been possible. On the other hand, microbial resources are the material basis of the biotechnology industry, with wide-ranging applications in industries such as industry, agriculture, environment, food, and medicine. Therefore, the isolation and screening of microbial resources form the foundation for the development of biology and modern biotechnological techniques.

Since the discovery of microorganisms, scientists have been trying to isolate microbial strains. It was not until 1881 that German microbiologist Robert Koch established the technique of solid plate isolation for microbial pure cultures [4]. This technique has been widely used in microbial diversity analysis and the isolation and cultivation of pure cultures, providing the prerequisites for the development of various fields of microbiology, such as microbial taxonomy, physiology, genetics, etc. Subsequently, researchers discovered a phenomenon known as the “great plate count anomaly”, which refers to the difference between the total number of microorganisms and the number of cultivable microorganisms [5]. This indicates that the majority of microorganisms in nature are uncultivable. It is estimated that approximately 1011–1012 species of microorganisms exist in various biospheres on Earth [6]. Traditional cultivation methods have been used for over 150 years to isolate microorganisms, resulting in the isolation of thousands of species. However, the currently cultivable microorganisms only account for 0.1–1.0% of the total microbial population [7]. Most microorganisms cannot be cultured and are often referred to as “microbial dark matter”. In recent years, with the development of molecular biology techniques and bioinformatics, non-culture-dependent methods such as denaturing gradient gel electrophoresis (DGGE), amplicon sequencing, and metagenomic sequencing have become powerful tools for revealing the abundance and functions of species within microbial communities [8,9,10]. However, it is necessary to obtain pure cultures to explore the metabolic characteristics and potential functions of microbial strains.

## 3. The Reason Why There Are So Many Uncultured Microorganisms in Nature

### 3.1. Most Uncultured Microorganisms Exist in a Viable but Non-Culturable State

In nature, the majority of uncultured or difficult-to-culture microorganisms exist in a dormant form, known as the viable but non-culturable (VBNC) state. The VBNC state was first discovered by Xu and Colwell in 1982 during their study of *Vibrio cholerae* and *Escherichia coli* survival in marine and estuarine environments [11]. The VBNC state is a survival strategy employed by bacteria in response to unfavorable growth conditions. Bacteria in the VBNC state cannot grow on conventional culture media but remain viable and maintain low metabolic activity. It has been found that various conditions, such as an unsuitable culture temperature, pH, nutrient limitation, and antibiotic stress, can induce microorganisms to enter the VBNC state [12,13,14]. Over 100 microbial species have been discovered to be capable of entering the VBNC state, with the majority being bacteria, along with a smaller number of archaea and fungi [15]. The formation of the VBNC state is complex, and its underlying mechanisms are not yet fully understood. The widely accepted “gene regulation theory” suggests that the VBNC state is a survival strategy employed by non-spore-forming bacteria, and its formation is regulated by genes, similar to the spore formation mechanism [16,17]. Extensive research has been conducted on the proteins and regulatory pathways involved in the formation of the VBNC state, including the stringent response, toxin–antitoxin (TA) systems, and RNA polymerase RpoS, which are believed to play a role [18].

### 3.2. Inability of Laboratories to Simulate the Native Natural Environment for Microbial Growth

The natural environment is complex and ever-changing, and our understanding of the microbial growth environment is not sufficiently deep. As a result, laboratories cannot fully simulate the native ecological environment for microbial growth. For example, there is a severe lack of knowledge about the essential nutrients, small molecular signaling molecules, nutrient requirements at different physiological stages, and patterns of changes in the growth environment conditions required for microbial growth. Under laboratory conditions, it is impossible to provide ideal growth conditions for microbial growth, leading to failures in isolating and culturing target microorganisms [19]. Additionally, the natural environment for microbial survival is diverse and includes various extreme environments, such as high/low temperature, high/low salinity, high/low alkalinity, anaerobic conditions, polar glaciers, permafrost, saline–alkaline soil, and the seafloor. Although scientists have made efforts to detect various indicators of extreme environments and isolate microbial populations from them, the current experimental conditions and methods still cannot fully simulate the growth conditions of microorganisms in the natural environment. Therefore, researchers often design isolation and culture methods based on the major environmental factors while neglecting many uncultured or difficult-to-culture microbial groups in extreme growth conditions [20,21].

### 3.3. Low Abundance and Weak Competitiveness

In the natural environment, there are many microorganisms with low abundance but strong metabolic functions. However, scientists still cannot determine the optimal cultivation methods for these low-abundance microorganisms, leading to the ineffective enrichment of these low-abundance bacterial strains [19], and thus, the inability to obtain pure cultures of these low-abundance microorganisms. Moreover, there are microbial groups in the natural environment that have relatively slow growth rates. Under certain isolation and cultivation conditions, fast-growing microorganisms reproduce in large numbers, occupying a significant portion of the growth resources and living space, thereby suppressing the reproduction of slow-growing microorganisms. Traditional isolation and cultivation methods have difficulty selecting slow-growing microbial groups. These microorganisms exist in low abundance in the natural environment and cannot be detected using methods such as amplicon sequencing, which hinders the simulation of the growth conditions for these low-abundance microorganisms [20].

### 3.4. Neglect of Microbial Interactions in the Environment

The natural environment is home to diverse microbial species, including bacteria, actinomycetes, cyanobacteria, chlamydia, rickettsia, protozoa, fungi, and viruses. They do not exist in isolation but are part of a complex network of interactions. The complex relationships within these microbial communities impact the isolation and cultivation of microorganisms, including parasitism, mutualism, commensalism, competition, and antagonism [22,23,24]. The microbial communities in these systems cooperate and depend on each other, with the growth of one organism relying on nutrients secreted by another, such as amino acids, bases, or vitamins [25]. One microorganism may influence the growth and reproduction of another. However, traditional isolation and cultivation methods disrupt these interdependencies, leading to failures in isolating and culturing the target microorganisms [21]. Furthermore, there is limited research on the highly complex interactions among environmental microorganisms, resulting in insufficient understanding to guide the isolation of uncultured microorganisms.

### 3.5. Inability to Adapt to Drastic Changes in Nutrient Environments

Organisms in nature have evolved and adapted to their natural environment over long periods of time. Microorganisms have developed their own growth and survival strategies in specific natural environments. Some microorganisms have chosen strategies of rapid growth, low survival rates, and reliance on high reproduction rates. Others have adapted to low nutrient levels, with a high affinity for environmental resources but extremely low growth rates [26]. When attempting isolation and cultivation, microorganisms often struggle to adapt to drastic changes in cultivation environments and nutrient conditions, resulting in their inability to grow and reproduce on culture media. Most microorganisms in their natural conditions exist in oligotrophic states. When cultured on nutrient-rich media, fast-growing and resistant microorganisms can quickly multiply, depleting the nutrients and inhibiting the growth of slow-growing or less-resistant microorganisms. Additionally, some oligotrophic or slow-growing microorganisms may form small colonies on culture media, making them difficult to isolate due to the limitations of detection methods or the incubation time [27].

## 4. Uncultured Microorganisms in the Fermentation Systems of Traditional Chinese Fermented Foods

### 4.1. Traditional Chinese Fermented Foods and Their Microorganisms

China has a wide variety of food ingredients and exhibits diversity in the traditional fermented foods. They mainly include alcoholic beverages such as Baijiu (Chinese liquor), Huangjiu (yellow rice wine), and rice wine; soybean sauce such as soy products, fermented black beans, soybean paste, and fermented tofu; cereal products such as aged vinegar; dairy products such as yogurt and milk tofu; as well as sauerkraut and sausages. The fermentation systems of traditional Chinese fermented foods typically involve the fermentation of multiple strains of microorganisms, resulting in rich microbial diversity. Through continuous selection and enrichment in the surrounding environment and during the fermentation process, a unique microbial community has been established in the fermentation systems of traditional Chinese fermented foods [28]. The specific statistical data are shown in Table 1. In addition to the dominant microorganisms listed in Table 1, *Archaea* are also dominant microorganisms in Baijiu cellar mud. They mainly include *Methanoculleus*, *Methanobrevibacter*, *Methanosarcina*, *Methanocorpusculum*, *Methanobacterium*, etc. [29]. Studies have shown that *Archaea* play a prominent role in increasing the content of flavor compounds, such as ethyl caproate, in strong aroma-type Baijiu [30].

### 4.2. Uncultured Microorganisms in Traditional Chinese Fermented Foods

The fermentation systems of traditional Chinese fermented foods contain abundant microorganisms, making them a vast repository of microbial resources. However, the majority of these microorganisms are currently difficult to culture or unculturable. Researchers have already discovered the presence of uncultured microorganisms in traditional Chinese fermented foods such as pickled vegetables [54] and cured ham [55]. Some of these unculturable microorganisms have been found to have high abundance within the fermentation systems and to represent dominant species within the systems [55]. Yun Jia et al. [56] isolated five strains of microorganisms (*Aspergillus oryzae*, *Bacillus subtilis*, *Staphylococcus gallinarum*, *Weissella confusa*, *Zygosaccharomyces rouxii*) from broad bean paste as core species to reconstruct a synthetic microbial community, which was then applied in the fermentation of broad bean paste. Compared to traditionally naturally fermented broad bean paste, the flavor compounds in the synthetic community were similar in terms of the types of compounds present but showed differences in their concentrations. The alcohol and aldehyde compounds were reduced, while the esters and phenolic compounds were increased. This may indicate that the uncultured microorganisms within natural fermentation systems play a regulatory role in determining the types and concentrations of flavor compounds during the fermentation process. Therefore, it is necessary to explore more microbial resources, optimize the composition of artificial microbial communities, and ultimately, achieve a microbial community that closely resembles natural fermentation systems. Additionally, previous studies have successfully isolated uncultured microbial species from traditional Chinese fermented foods and their fermentation systems, such as Baijiu cellar mud (as shown in Table 2), further confirming the existence of uncultured microorganisms within fermentation systems.

Research on the uncultured microorganisms in fermented foods can increase the cultivable quantity of microorganisms, which is beneficial for the further development of microbial products and the better utilization of microorganisms in human production and daily life.

## 5. Research Methods for Uncultured Microorganisms in the Fermentation Systems of Traditional Fermented Foods

### 5.1. Optimization of Cultivation Methods and Culture Media

By improving the cultivation conditions, many previously unculturable microorganisms have been successfully isolated and cultured. It is highly feasible to apply this approach to the cultivation of difficult-to-culture microorganisms in traditional fermented foods’ fermentation systems

#### 5.1.1. Enrichment Cultivation

Enrichment cultivation is a complex mixed-culture system. Studies have shown that enrichment cultivation can increase the abundance of certain microorganisms and revive dormant microorganisms, thereby increasing the number and diversity of cultivable microbial [83]. Most traditional Chinese fermented foods undergo mixed-culture natural fermentation, where microorganisms interact with each other [1] and some microorganisms may be dormant or in a viable but non-culturable (VBNC) state. Therefore, pre-enrichment cultivation before pure cultivation may help revive the abundance of some unculturable microorganisms. Chao Lan Liu et al. [84] designed an enrichment system to enrich microorganisms from Baijiu cellar mud, and the enrichment system DSMZ 827 was designed based on the growth conditions of a dominant bacterium, *Petrimonas sulfuriphila* BN3, found in the pit mud. The composition of the enrichment medium was retrieved from the Microbial Culture Collection Database of the DSMZ (German Collection of Microorganisms and Cell Cultures) (https://www.dsmz.de/collection/catalogue, accessed on 22 January 2023), which recommends specific culture media based on microbial species. This study successfully isolated and cultured an anaerobic potential new species after enrichment. Additionally, it has been demonstrated that using low-nutrient enrichment media with rapidly utilizable carbon and limited nitrogen sources promotes the growth and revival of slow-growing oligotrophic bacteria (k-strategists) and dormant microorganisms in oligotrophic environments [85]. In the fermentation system, Baijiu cellar mud is exposed in anaerobic, weakly acidic, and oligotrophic conditions for a long period [86]. Lu Mengmeng [87] enriched microorganisms from Baijiu cellar mud by using three enrichment media with progressively reduced nutrient components. The results showed that the low-nutrient medium with the lowest concentrations of readily available carbon and nitrogen sources yielded the best enrichment results, with the highest microbial diversity, while the nutrient-rich medium resulted in a more monotonous microbial population. The design of this enrichment medium was based on the growth conditions of Clostridium acetobutylicum ATCC 824 [88]. Furthermore, substances that promote the revival of VBNC microorganisms, such as sodium pyruvate, can be added to the enrichment culture medium to achieve the pure cultivation of some previously unculturable microorganisms [85].

#### 5.1.2. Improved Culture Media

Researchers have designed culture media specific to the growth conditions of microorganisms to isolate more difficult-to-culture microorganisms. Han Ying [89] designed a culture medium using the liquid obtained from dissolving and centrifuging Baijiu cellar mud and fermentation mash instead of distilled water to better simulate the in situ environment. Gao Feng [90] used the characteristic organic acid salts produced during the fermentation of Northeast sauerkraut as specific carbon and nitrogen sources in the culture medium to isolate strains that could decompose harmful organic acids in the sauerkraut fermentation liquid and obtained three new bacterial species. Liu Wenrong [91] added metavanadate, a growth factor that promotes the growth of spoilage microorganisms on conventional culture media, to the standard MRS culture medium, which allowed for the first isolation of the key microorganisms responsible for acid spoilage in aged Huangjiu (yellow rice wine). The modified culture media better meet the growth requirements of difficult-to-culture microorganisms and enable the cultivation of microorganisms that cannot grow under conventional culture conditions.

### 5.2. Integration of High-Throughput Sequencing Technology and Pure Cultivation Methods

With the development of sequencing technology and bioinformatics analysis, high-throughput sequencing techniques, such as metagenomics and amplicon sequencing, have become powerful tools for revealing the abundance and functional potential of microbial communities [92]. Moreover, antiSMASH was used to identify the secondary metabolite gene cluster among shotgun metagenomics. However, to explore the metabolic characteristics of strains and their roles in the fermentation process, obtaining pure cultures is necessary. By combining culture-independent high-throughput sequencing methods with traditional cultivation methods, both approaches can complement and validate each other, providing a more comprehensive and in-depth understanding of microbial diversity. Yi Fan et al. [93] compared the results of culture-dependent and metagenomic sequencing methods and found that the *Lactobacillus* sp. and *Saccharomyces cerevisiae* in Fenjiu Daqu (fermentation starter for Chinese liquor) could only be detected using culture-independent methods. Gao Feng [90] studied the bacterial diversity in pickled cabbage fermentation samples at different stages using high-throughput sequencing technology and combined it with innovative culture media to isolate new bacterial species. Liu Wenrong [91] used high-throughput sequencing technology in combination with traditional cultivation methods to study the major microbial species causing acid spoilage in aged Huangjiu, successfully isolating two key bacterial strains responsible for acid spoilage for the first time.

### 5.3. Metabolomics

Metabolomics is a systematic analysis of metabolites and their temporal changes in biological tissues. It mainly studies endogenous and exogenous small molecules, such as peptides, amino acids, carbohydrates, and fatty acids, reflecting the biochemical status of cells [94]. Metabolomics can be divided into targeted and untargeted analyses, with the commonly used research methods including liquid chromatography–tandem mass spectrometry (LC–MS/MS), high-performance liquid chromatography–time-of-flight mass spectrometry (HPLC–TOF/MS), nuclear magnetic resonance (NMR), gas chromatography–mass spectrometry (GC–MS), etc. [95]. In the field of fermented foods, Guiliang Tan et al. [47] used a combination of high-throughput sequencing and metabolomics to study the correlation between the bacterial community diversity and metabolites in Hongfu tofu and Baijifu tofu. This study analyzed the role of microbial communities in the production of metabolites in tofu, providing a theoretical basis for the industrial utilization of different functional microbial species. After analyzing the correlation between the microorganisms and organic substances, it is possible to consider adding exogenous metabolites as growth substrates during cultivation to achieve the isolation and cultivation of unculturable microorganisms. Lu Mengmeng [87] added acetic acid, butyric acid, and formic acid–butyric acid, which are metabolic substrates for microorganisms based on the metabolic characteristics of the main microbial communities in Baijiu cellar mud, to the enrichment culture medium. The dynamic changes in the content of the substrates (short- and medium-chain fatty acids) and products (methane) were monitored during the enrichment cultivation process, and the results showed that with the consumption of the substrates, the content of the products increased. The low-abundance microorganisms in the cellar mud were enriched in all three groups of enrichment media, providing a new approach for the cultivation of uncultured microorganisms in Baijiu cellar mud.

### 5.4. Culturomics

Obtaining pure cultures of microorganisms solely through cultivation-independent methods is challenging and hinders the development and utilization of microbial resources. Cultivation-dependent methods are indispensable for studying microbial functions, although using a single culture condition to isolate microorganisms has low efficiency. Therefore, the field of culturomics has emerged. Culturomics, a technology for culturing microorganisms created by Lagier et al. in 2012, involves designing multiple cultivation and selection conditions to promote the growth of low-abundance or difficult-to-culture microorganisms while inhibiting the growth of most populations. The isolated bacteria are identified at the species level using matrix-assisted laser desorption/ionization time-of-flight mass spectrometry (MALDI–TOF-MS) and 16S rRNA gene sequencing [96]. Culturomics has the potential to screen low-abundance microorganisms and complement the limitations of traditional cultivation techniques. In the field of traditional fermented foods, Jialiang Xu et al. [97] used a combination of culturomics and amplicon sequencing techniques to cultivate bacteria from Baijiu cellar mud. Based on the amplicon sequencing results, they designed culture media and conditions suitable for the survival of different microbial species in cellar mud, including commercial culture media, simulated in situ environmental culture media, and predicted the culture media retrieved from the Known Media Database (KOMODO) (http://komodo.modelseed.org, accessed on 20 January 2023). Through culturomics, a total of 215 bacterial pure cultures were obtained, belonging to 41 genera in 4 phyla, with 36 genera not previously isolated from cellar mud. This study also cultivated 19 potential new species, significantly increasing the number of cultivable microorganisms from Baijiu cellar mud. It was the first time that culturomics was applied to the isolation of microorganisms from Baijiu fermentation systems, enriching the pure culture library of cellar mud microorganisms and holding important significance for studying uncultured microorganisms in Baijiu cellar mud. The flavor compound 3-(methylthio)-1-propanol is present in Baijiu, and Du Rubing et al. [98] developed a synthetic microbial consortium based on the division of labor of multiple modules. They isolated the core strains (yeast, spore-forming bacteria, lactic acid bacteria) for each module through culturomics and used them to achieve the efficient biosynthesis of 3-(methylthio)-1-propanol. They designed selective culture media and screened the target strains by adding antibiotics, observing the formation of transparent hydrolysis circles, and using color changes in the media. However, it should be noted that culturomics requires the design of multiple cultivation conditions to meet the preferences of different species in microbial communities, while the microbial community compositions may vary in different samples. Therefore, there are currently limitations in terms of the workload and time requirements. In the field of gut microbiota applications, Yuxiao Chang et al. [99] employed methods such as extending the enrichment culture time, adding fresh culture medium after sampling, sampling at representative time points, and selecting “experienced” colonies (i.e., selecting representative colonies for isolation and purification) to increase the number of cultivable microorganisms while reducing the workload. Fengyi Hou et al. [100] added CHIR-090, an inhibitor of the rapid growth of Gram-negative bacteria such as *E. coli* and *P. aeruginosa*, to the enrichment medium to create growth space for other microbial populations and isolated new species from fresh fecal samples. It is possible to optimize culturomics conditions based on the relative abundance characteristics of the microorganisms in the fermentation system or to selectively isolate strains with specific functions. In the future, culturomics will play an important role in obtaining pure cultures of uncultured microorganisms.

### 5.5. Inducing the Resuscitation of VBNC Microorganisms

When microorganisms are exposed to unfavorable environmental conditions, such as low temperature, desiccation, low pH, or nutrient depletion, they enter a viable but non-culturable state (VBNC). Microorganisms in the VBNC state cannot grow on conventional culture media but still retain certain activity [11]. Many microorganisms in the natural environment are non-culturable in the VBNC state, which is one of the important reasons why the majority of microorganisms in nature remain uncultivated. The VBNC state is a survival strategy of bacteria in response to adverse growth conditions, where bacteria in the VBNC state cannot grow on conventional culture media but still maintain viability and low metabolic activity.

The microbial diversity within the brewing systems of traditional Chinese fermented foods is extremely complex. Taking strong-flavored Baijiu as an example, the centuries-old fermentation starters and cellars have undergone continuous selection and enrichment through the surrounding environment and fermentation processes, forming a unique microbial community and a vast reservoir of microbial resources. However, the fermentation starters, cellars, fermentation mash, and surrounding environment often experience unfavorable conditions such as nutrient depletion, high temperature, desiccation, high acidity, and high ethanol content, which may lead to the entry of many microorganisms into the VBNC state.

VBNC microorganisms can be resuscitated into cultivable states under suitable conditions. Currently, the proven methods for resuscitation include improving culture conditions/providing suitable growth conditions, eliminating stressors, adding organic compounds such as sodium pyruvate, adding oxidoreductases, and adding resuscitation promoting factor (Rpf). Rpf is a protein that was first isolated from *Micrococcus luteus* and has the ability to resuscitate VBNC microorganisms [101,102]. Telkov et al. found that the Rpf protein from *Micrococcus luteus* has peptidoglycan hydrolase activity [103]. It consists of 220 amino acid residues, with 75 residues forming a conserved region at the N-terminus, which constitutes a lysozyme-like fold. The C-terminus consists of 109 amino acid residues, forming a variable region that includes an LysM domain, which may help the Rpf protein to locate and enhance its lytic activity on substrates [104]. Genes encoding Rpf-like proteins are widely present in high-GC content Gram-positive bacteria. The specific mechanism of Rpf-mediated resuscitation is not yet clear but mainly exists in two possible models. The first model suggests that Rpf directly acts on the peptidoglycan components in the cell walls of VBNC bacteria, inhibiting cell division or growth and causing them to recover to normal bacterial states. The second model suggests that after Rpf degrades peptidoglycan, certain products act as “second messengers” that interact with other factors, promoting the growth of VBNC bacteria [105].

Wang Jiaxuan isolated two new strains, *Umezawaea beigongshangensis* REN6 and *Corynebacterium beijingensis* REN39, from Baijiu cellar sediments. Two Rpf genes, Rpf-R62 and Rpf-R39, were cloned from these strains, both of which encode proteins with the ability to promote the resuscitation of VBNC bacteria and stimulate the growth of Acinetobacter and Bacillus bacteria. The Rpf-R39 protein exhibited good growth-promoting effects on Brevundimonas and Acinetobacter bacteria [106]. The Rpf proteins can promote the growth of specific bacterial genera, thereby affecting the abundance and diversity of the microbial communities in cellar sediments. In addition, the results of pure cultivation methods showed that the addition of the two Rpf proteins to the enrichment cultures significantly increased the number of bacterial genera and species isolated from the sediments. Furthermore, the addition of the Rpf-R62 protein led to the isolation of a strain suspected to be a new species.

### 5.6. Extending the Cultivation Time

Many microorganisms, especially those living in nutrient-limited environments, have slow growth rates. Therefore, extending the cultivation time can significantly improve the cultivation rate of uncultured microorganisms, facilitating the growth of slow-growing microorganisms. Pulschen et al. successfully isolated rare microorganisms such as *Lapillicoccus*, *Flavitalea*, *Quadrisphaera*, *Motilibacter*, and *Polymorphobacter* from Antarctic soil samples by using low-nutrient media, extending the cultivation time, and selectively cultivating slow-growing bacteria [107]. Choi et al. isolated two new marine bacteria, *Mooreiaceae* and *Catalimonadaceae*, from marine sediments by using low-nutrient media and extending the cultivation time [108]. He collected various types of samples from polar environments and improved the cultivation temperature and extended the cultivation time, successfully isolating approximately 1220 bacterial strains, including 182 potential new species [109]. Sun et al. successfully isolated a new anaerobic bacterium, *Flexilinea flocculi gen. nov*. sp. nov., from sludge in a medium-temperature upflow anaerobic sludge bed reactor by using enriched culture media and extending the cultivation time to 14 days [110].

### 5.7. Co-Cultivation

Microorganisms in the natural environment do not exist in isolation but are part of a complex ecological interaction network. Many bacteria in the natural environment rely on symbiotic relationships for growth, where the growth of one microorganism may provide nutritional substances and signaling molecules to others. Microbial interactions enable multifunctional cultivation that cannot be achieved through the cultivation of individual species alone. For example, a combination of mixed cells can accomplish multi-step cellulose degradation, while intermediate products can be utilized as carbon sources [111]. Researchers have improved cultivation rates by adding auxiliary bacterial strains during co-cultivation. For instance, Nichols et al. successfully isolated a novel psychrophilic genus of bacteria by adding auxiliary strains from enriched bacterial communities [112]. Wu used co-cultivation followed by flow cytometric cell sorting after 30 days to isolate various bacteria, including 8 potential new species [113]. Xian et al. analyzed the metagenomes of microbial communities in hot springs in Yunnan and Tibet, unraveling the complex interaction networks among environmental microorganisms. Through co-cultivation, they obtained a large number of uncultured bacterial strains belonging to the phylum Chloroflexi, including 36 potential new species [114].

### 5.8. Microfluidic Cultivation Techniques

Microfluidic cultivation techniques involve the control, manipulation, and detection of complex fluids at the microscale level. These techniques allow for the simultaneous detection of multiple uncultured microorganisms in a short period and for the acquisition of pure cultures of target microorganisms [115]. Microfluidic cultivation and analysis devices are powerful tools for studying microbial interactions at the single-cell level. Compared to traditional cultivation methods, this technique entails complex manufacturing processes and specialized equipment. Jiang et al. developed a microfluidic streak plate method (MSP) for isolating many uncultured microorganisms from soil samples enriched with polycyclic aromatic hydrocarbons, including a new species of Blastococcus capable of fluoranthene degradation [116]. Zhao et al. designed and fabricated a microfluidic chip consisting of individual elements with U-shaped structures and grooves on the upper part. This chip captured single fungal cells, enabling their cultivation for over 48 h and allowing the observation of budding and spore germination processes. The chip was used for the capture, cultivation, localization, and real-time observation of single fungal cells, offering simplicity, convenience, and intuitive operation [117]. Zhang et al. used microfluidic technology to sort antibacterial-active actinobacteria, successfully demonstrating its feasibility, which distinguished it from traditional screening methods and provided advantages such as increased efficiency and reduced cultivation time [118]. Microfluidic techniques can overcome some limitations of traditional pure culture methods, reduce interspecies competition, and ensure the growth of low-abundance and slow-growing microorganisms. However, microfluidic technology requires complex and specialized equipment, making it more expensive to isolate microorganisms.

### 5.9. Cultivation Techniques Based on Cell Sorting

Cell sorting techniques are commonly used to isolate individual cells from complex microbial communities for subsequent cultivation. Methods such as optical tweezers, fluorescence in situ hybridization, fluorescence-activated cell sorting, Raman-activated single-cell sorting (RACS), and reverse genomics for targeted isolation are representative cell sorting techniques, which are often combined with microfluidic technology. Irie et al. reported a novel method for physically enriching uncultured acidophilic bacteria and nitrifying bacteria from activated sludge microbial communities using a cell sorting system [119]. Abe et al. reported a high-throughput separation technique based on scattering characteristics, which directly isolated ammonia-oxidizing bacterial colonies from activated sludge, avoiding long-term cultivation and enrichment biases. The isolated colonies were then inoculated into culture media in 96-well plates for the successful isolation of novel strains within the Nitrosomonas genus [120]. Cross et al. used cell sorting techniques to isolate uncultured microorganisms from human oral samples, resulting in the isolation of two strains [121]. By analyzing the genomic data of the target strains, performing a reverse analysis, and studying the binding antibodies of the target microorganisms, they applied this approach to environmental samples, enabling interaction with the target microorganisms. Subsequently, the target microorganisms were isolated using fluorescence-activated cell sorting technology. Fujitani et al. used a combination of forward scatter and side scatter to isolate three uncultured strains of Nitrospirae from enriched samples [122]. These methods enable the precise isolation of individual cells, providing both cultivation conditions and the elimination of inter-species competition. However, these techniques have lower output and shorter establishment times, and their technical maturity still needs further exploration.

## 6. Exploitation and Utilization of Uncultured Microorganisms in Traditional Fermented Foods’ Fermentation Systems

### 6.1. Screening of Microorganisms with Specific Functions for Development and Utilization

The microbial resources in traditional fermented foods’ fermentation systems are abundant and play an important role in the fermentation process. Microorganisms with specific functions can be screened from these resources for development and utilization, and microbial preparations can also be prepared for applications in other production fields. For example, researchers have isolated a new species of halophilic bacteria, Lentibacillus panjinensis, which is tolerant to high salt concentrations, from shrimp paste. In addition, a new species capable of efficiently producing butyric acid using lactate has been discovered in Baijiu cellar mud. The specific statistics are shown in Table 3.

### 6.2. Artificial Construction of Core Microbial Consortia

Studying the metabolic pathways and functions of uncultured microorganisms during the fermentation process and investigating the impacts of their metabolites on the fermentation process and flavor of fermented foods is helpful for analyzing the core microbes required for the construction of artificial fermentation microbial consortia. Taking Baijiu (Chinese liquor) as an example of a traditional fermented food, Qun Wu et al. [131] pointed out that the focus of Baijiu fermentation is to study the microbial community that produces the flavor compounds and factors that influence the core microbial community and to isolate the key microbial community related to flavor compound production. By isolating the uncultured microorganisms in traditional fermented foods and determining whether they are key microorganisms, their metabolic pathways can be studied to elucidate the fermentation mechanisms and achieve the goal of artificially constructing core microbial consortia. Yongjian Yu et al. [132] obtained a new species of lactic acid bacteria named Acetilactobacillus jinshanensis from the vinegar mash of traditional Zhenjiang aromatic vinegar. Subsequently, Sun Jia [133] analyzed the growth characteristics and functional potential of Acetilactobacillus jinshanensis and conducted pilot experiments to explore its functions in vinegar brewing. The results showed that Acetilactobacillus jinshanensis could produce lactic acid, acetic acid, and citric acid using sugar substances in the culture medium. In the brewing experiment, the addition of Acetilactobacillus jinshanensis at the initial stage of the acetic acid fermentation accelerated the increase in the fermentation system’s temperature and glucose consumption rate, thereby shortening the fermentation cycle. At the end of the acetic acid fermentation, the physicochemical indicators of vinegar mash were favorable and the content of some beneficial volatile compounds significantly increased. Further exploration will be conducted concerning its application in grain vinegar and other fermented foods. Xu Jialiang [123] isolated a new bacterial species called Pontibacter beigongshangensis from the Maiqu (wheat starter) of Beizong Huangjiu (a type of Chinese yellow rice wine). Shuyue Hao [124] found that Pontibacter beigongshangensis induced the degradation of biogenic amines in Huangjiu by Pediococcus pentosaceus M28. Li Yihan et al. [125] further studied the pigment composition of Pontibacter beigongshangensis and found that it contained carotenoid-like compounds, which were stable under strong acid and ultraviolet conditions but unstable under strong alkali and high temperature conditions. The discovery of these new bacterial species provides more options for the artificial construction of core microbial consortia in fermentation systems.

## 7. Conclusions and Prospects

There are still many microbial resources in traditional fermented foods and their fermentation systems in China that require further exploration and research. The cultivation of difficult-to-culture microorganisms is crucial, as only a small proportion of microorganisms can be cultivated in the natural environment. Currently, there are a diversity of methods for cultivating difficult-to-culture microorganisms, such as co-cultivation based on microbial interactions, diffusion chambers, isolation chips, and other specific devices that place samples in their original environments for in situ cultivation. Reverse genomics, extinction dilution cultivation, and other methods have also achieved practical success in different fields. Therefore, (1) in situ cultivation and co-cultivation techniques can be used to isolate microorganisms from fermentation systems; (2) the combination of cultivation genomics and extinction dilution can allow microorganisms to grow in trace amounts, reducing competition for nutrients, while the diverse culture conditions designed by cultivation genomics can satisfy the growth requirements of more microorganisms; and (3) exploring the relationship between fermentation system microorganisms and the viable but nonculturable (VBNC) state, and verifying and resuscitating VBNC microorganisms in fermentation systems by adding resuscitation-promoting factors, etc., may become important means of studying uncultured microorganisms in traditional Chinese fermented foods.

Studying, isolating, and exploiting uncultured microorganisms in fermented foods and fermentation systems using various techniques contribute to the understanding of the microbial diversity in fermentation systems and the relationship between microorganisms and metabolites during fermentation. This promotes the mature, improved, and industrial development of traditional Chinese fermented food fermentation processes while maintaining their unique Chinese flavors. Additionally, it increases the number of cultivable microorganisms, providing more resources for the industrial development and application of microbial strains.

## Figures and Tables

**Table 1 foods-12-02691-t001:** Microorganisms in traditional Chinese fermented foods’ fermentation systems.

Traditional Chinese Fermented Foods	Fermentation Systems	Predominant Microorganisms		References
		Fungi	Bacteria	
Baijiu	Cellar mud	*Saccharomycopsis*, *Candida*, *Penicillium*, *Aspergillus*, *Pichia*, *Mortierella*, *Pseudeurotium*, *Byssochlamys*, *Scedosporium*	*Clostridium*, *Bacillus*, *Lysinibacillus*, *Hydrogenispora*, *Paenibacillus*, *Acinetobacter*, *Petrimonas*, *Pediococcus*, *Syntrophomonas*, *Ruminococcus*, *Sporosarcina*, *Sedimentibacter*	[31,32,33,34]
	Fermented grains	*Saccharomyces*, *Pichia*, *Saccharomycopsis*, *Thermoascus*, *Candida*, *Aspergillus*, *Rhizomucor*, *Monascus*, *Byssochlamys*	*Lactobacillus*, *Kroppenstedtia*, *Bacillus*, *Lactococcus*, *Weissella*, *Acetobacter*, *Thermoleophilum*	[33,34,35]
	Daqu	*Aspergillus*, *Candida*, *Rhizopus*, *Rhizomucor*, *Saccharomycopsis*, *Monascus*	*Lactobacillus*, *Weissella*, *Bacillus*, *Lentibacillus*, *Thermoactinomyces*, *Kroppenstedtia*, *Saccharopolyspora*	[36,37,38,39]
	Xiaoqu	*Rhizopus*, *Aspergillus*, *Candida*, *Wallemia*, *Monascus*, *Xeromyces*, *Saccharomyces*	*Staphylococcus*, *Weissella*, *Lactobacillus*, *Bacillus*, *Enterobacter*, *Acinetobacter*, *Corynebacterium*	[40,41]
Aged vinegar	Cupei	*Fusarium*, *Alternaria*, *Epicoccum*, *Aspergillus*, *Saccharomyces*, *Eurotium*	*Lactobacillus*, *Acetobacter*, *Bacillus*, *Acaryochloris*, *Acetilactobacillus*, *Agrobacterium*, *Sphingomonas*, *Weissella*, *Gluconacetobacter*	[42,43]
Cheese		*Geotrichum*, *Candida*, *Pichia*, *Saccharomyces*, *Rhodotorula*	*Streptococcus*, *Lactobacillus*, *Lactococcus*, *Acinetobacter*, *Enterococcus*	[44,45,46]
Fermented tofu		*Aspergillus*, *Rhizopus*, *Mucor*, *Pichia*, *Candida*, *Guehomyces*, *Schizosaccharomyces*	*Lactococcus*, *Acinetobacter*, *Tetragenococcus*, *Lactobacillus*, *Enterococcus*, *Pseudomonas*	[47,48]
Sauerkraut		*Candida*, *Pichia*, *Saccharomyces*, *Fusarium*	*Lactobacillus*, *Pediococcus*, *Weissella*, *Lactococcus*, *Enterobacter*	[49,50,51]
Sausages		*Aspergillus*, *Candida*, *Debaryomyces*, *Lachancea*, *Millerozyma*, *Schwanniomyces*	*Lactobacillus*, *Acinetobacter*, *Lactococcus*, *Macrococcus*, *Serratia*, *Weissella*	[52,53]

**Table 2 foods-12-02691-t002:** New microbial species isolated from traditional Chinese fermented foods’ fermentation systems.

Traditional Chinese Fermented Food Fermentation System	Genus of New Species	References
Cellar mud	*Sporosarcina*, *Aminipila*, *Lysobacter*, *Bacillus*, *Pseudoxanthomonas*, *Clostridium*, *Blautia*, *Paenibacillus*, *Novisyntrophococcus*	[57,58,59,60,61,62,63,64,65,66]
Daqu	*Brevibacterium*, *Thermoflavimicrobium*, *Franconibacter*, *Scopulibacillus*	[67,68,69,70]
Sauerkraut	*Lactobacillus*, *Enterococcus*, *Yarrowia*	[71,72,73]
Yogurt	*Weissella*, *Lactobacillus*, *Enterococcus*, *Leuconostoc*	[74,75,76,77]
Fermented tea	*Aeromicrobium*, *Pullulanibacillus*, *Paenibacillus*, *Pueribacillus*, *Bacillus*	[78,79,80,81,82]

**Table 3 foods-12-02691-t003:** Uncultured microorganisms with specific functions in traditional Chinese fermented foods’ fermentation systems.

Traditional Chinese Fermented Foods’ Fermentation System	Strains	Functions	References
Wheat Qu	*Pontibacter beigongshangensis*	Produce pigment, induce *Pediococcus* to produce bacteriostatic substance and reduce the content of biogenic amine in Huangjiu	[123,124,125]
Shrimp paste	*Lentibacillus panjinensis*	Halophilic, tolerant to 25% salt concentration	[126]
Cellar mud	*Clostridium* sp. BPY5	Produce lactic acid from butyric acid	[127]
Cellar mud	*Ruminococcaceae* bacterium CPB6	High yield of caproic acid from lactic acid	[128]
Fermented yogurt	*Bacillus* sp. DU-106	Produce L-lactic acid, with potential probiotic properties	[129]
Fermented tofu	*Bacillus* sp. ZJ1502	An alkaline protease capable of producing hydrogen peroxide resistance	[130]

## Data Availability

The data used to support the findings of this study can be made available by the corresponding author upon request.

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
