# Peer review of "Uncultured Microorganisms and Their Functions in the Fermentation Systems of Traditional Chinese Fermented Foods"

_foods, 2023, doi:10.3390/foods12142691_

Round 1

Reviewer 1 Report

The review article is well-written and organized. It needs minor corrections.

 1.    Italicize scientific names of the organism. Eg. L279: Escherichia coli, Pseudomonas aeruginosa

2.    L265: Remove highlight.

3.    Introduce the word before abbreviating. Eg: L278: CHIR

4.    Citations are not well formatted. Eg: L310-312: Y”ongjian Yu et al.[98] obtained…… Subsequently, Sun Jia [99] analyzed..”

Author Response

  1. Italicize scientific names of the organism. Eg. L279: Escherichia coli, Pseudomonas aeruginosa

Response: We had italicize the scientific names of the organism in the revised manuscript.

  1. L265: Remove highlight.

Response: Thanks for your question, this is a production name of a DNA inhibitor.

  1. Introduce the word before abbreviating. Eg: L278: CHIR

Response: Thanks for your question, this is a production name of a DNA inhibitor.

  1. Citations are not well formatted. Eg: L310-312: Y”ongjian Yu et al.[98] obtained…… Subsequently, Sun Jia [99] analyzed..”

Response: We had revised according to your comments.

Reviewer 2 Report

Introduction: line 28-line 35 change the citation according to author instructions e.g [1] to [1].

Please add more information in the introduction.

Line 86 the name of the Baiju must in italic.

Line 85 rewrite the sentences, “and, exhibit diversity in traditional fermented foods”.

Line 96 the word Archaea must in italic.

Line 97 the genus must in italic.

Line 120 what do you mean “artificial microbial communities”?

Line 122, rewrite the sentences to “previous studies have successfully isolated uncultured microbial species”.

Line 158 add “in anaerobic condition”.

Line 279 write in italic ( E. coli and P. aeruginosa).

Line 299-302: rewrite the sentences, as this sentence are not easy to follow by the reader.

the manuscript needs to reformat according to the instructions.

Author Response

  1. Introduction: line 28-line 35 change the citation according to author instructions e.g [1] to [1].

Response: We had revised the format of citation in the revised manuscript.

  1. Please add more information in the introduction.

Response: We had added more information in the introduction section.

  1. Line 86 the name of the Baiju must in italic.

Response: We had italic Baijiu in the revised manuscript.

  1. Line 85 rewrite the sentences, “and, exhibit diversity in traditional fermented foods”.

Response: We had rewrite the sentences.

  1. Line 96 the word Archaea must in italic.

Response: We had italic Archaea in the revised manuscript.

  1. Line 97 the genus must in italic.

Response: We had italic the genus name in the revised manuscript.

  1. Line 120 what do you mean “artificial microbial communities”?

Response: Thanks for your question. Artificial microbial communities means the microbial constitute in the community was known, and the microbes in the community was artificial added.

  1. Line 122, rewrite the sentences to “previous studies have successfully isolated uncultured microbial species”.

Response: We had rewrite the sentences.

  1. Line 158 add “in anaerobic condition”.

Response: We had added in the revised manuscript.

  1. Line 279 write in italic ( E. coli and P. aeruginosa).

Response: We had italic the genus name in the revised manuscript.

  1. Line 299-302: rewrite the sentences, as this sentence are not easy to follow by the reader.

Response: We had rewrite the sentences.

Reviewer 3 Report

Dear Authors,

The manuscript is a state-of-the-art review of the interesting topic of uncultivated microorganisms found in traditional Chinese fermented foods. You described what is the state of non-culturable, what are the reasons for it and you described the microorganisms that occur in Chinese fermented food. The chapter on research methods applied to uncultivated microorganisms and the use of these microbes in microbial consortia is very interesting. The work is written in understandable English, which requires minor corrections. References and their citations also need improvement. Please refer to the detailed notes below:

Title - the word "system" should be plural - "fermentation systems of...."

P2L87 you should put "soybean paste" after "soy sauce" (change the order of fermented products)

P3 L97-98 The names of microorganisms should be written in italics.

Tab.1. - in the header, correct "fungus" to "fungi" (plural as in bacteria).

Tab.3. "Functions" column - Shouldn't the name of the bacterium be Pediococcus pentosaceus instead of "pentose pediococcus"?

P9L337 - I didn't find citation of reference 100, perhaps that the first time given number 102 should be number 100? Please check it out.

P10L364-373 - reference instructions need to be removed, but please follow them before deleting (e.g. citation numbers should not be in superscript, no doi in references, I couldn't find citation number 19 in the text).

The work is written in understandable English, which requires minor corrections. I think that authors used gerund form too often

Author Response

  1. Title - the word "system" should be plural - "fermentation systems of...."

Response: We had revised the title according to your comments.

  1. P2 L87 you should put "soybean paste" after "soy sauce" (change the order of fermented products)

Response: We had revised according to your comments.

  1. P3 L97-98 The names of microorganisms should be written in italics.

Response: We had revised according to your comments.

  1. Tab.1. - in the header, correct "fungus" to "fungi" (plural as in bacteria).

Response: We had corrected according to your comments.

  1. Tab.3. "Functions" column - Shouldn't the name of the bacterium be Pediococcus pentosaceus instead of "pentose pediococcus"?

Response: We had revised according to your comments.

  1. P9 L337 - I didn't find citation of reference 100, perhaps that the first time given number 102 should be number 100? Please check it out.

Response: We had revised according to your comments.

  1. P10L364-373 - reference instructions need to be removed, but please follow them before deleting (e.g. citation numbers should not be in superscript, no doi in references, I couldn't find citation number 19 in the text).

Response: We had deleted the reference instructions according to your comments.

  1. The work is written in understandable English, which requires minor corrections. I think that authors used gerund form too often

Response: We had corrected the English language in the revised manuscript.

Reviewer 4 Report

In this review article, the authors tackle the issue of unculturable microorganisms in complex matrices and offer insight into their application in the industry. This review is concise and describes the most prevalent difficulties in the field while also detailing recent studies on the matter. Some specific points that require attention are:

Line 51: how is DGGE used as a culture-independent method to determine microbial abundance in complex samples?

Lines 145-147: is this method used to enrich a specific species or just to increase microbial abundance in a sample? A clarification should be added in the text.

Lines 195-200: there is a weak link between genomics (culture-independent identification), the capacity of strains to form biofilms, and isolation efficiency. Additionally, here the authors could also add information on shotgun metagenomics and their use to predict the metabolic capacity of the strains, and their substrate utilization patterns and thus provide information about appropriate culture conditions.

Lines 218-221: is there a direction of causality here? Does formic acid induce the proliferation of lactococci or does the increased abundance of lactococci lead to the accumulation of formic acid? Clarification can be made in the text.

Author Response

  1. Line 51: how is DGGE used as a culture-independent method to determine microbial abundance in complex samples?

Response: Thanks for your question. Because DGGE could distinguish DNA fragments with different bases constitute, therefore, in the complex samples, although the length of 16S rRNA was similar, their bases constitute was quit different, therefore, DGGE could be used as a culture-independent method to determine microbial abundance in complex samples.

  1. Lines 145-147: is this method used to enrich a specific species or just to increase microbial abundance in a sample? A clarification should be added in the text.

Response: Actually, this method could increase microbial abundance in a sample. We had revised in the revised manuscript.

  1. Lines 195-200: there is a weak link between genomics (culture-independent identification), the capacity of strains to form biofilms, and isolation efficiency. Additionally, here the authors could also add information on shotgun metagenomics and their use to predict the metabolic capacity of the strains, and their substrate utilization patterns and thus provide information about appropriate culture conditions.

Response: Thanks for your question, we are sorry for misunderstanding. We had revised this part according to your comments.

  1. Lines 218-221: is there a direction of causality here? Does formic acid induce the proliferation of lactococci or does the increased abundance of lactococci lead to the accumulation of formic acid? Clarification can be made in the text.

Response: Thanks for your question, we are sorry for misunderstanding. We had revised this part according to your comments.